# Towards a Combinatorial Characterization of Bounded-Memory Learning

**Alon Gonen**      **Shachar Lovett**      **Michal Moshkovitz**

University of California San Diego

## Abstract

Combinatorial dimensions play an important role in the theory of machine learning. For example, VC dimension characterizes PAC learning, SQ dimension characterizes weak learning with statistical queries, and Littlestone dimension characterizes online learning. In this paper we aim to develop combinatorial dimensions that characterize bounded memory learning. We propose a candidate solution for the case of realizable strong learning under a known distribution, based on the SQ dimension of neighboring distributions. We prove both upper and lower bounds for our candidate solution, that match in some regime of parameters. This is the first characterization of strong learning under space constraints in any regime. In this parameter regime there is an equivalence between bounded memory and SQ learning. We conjecture that our characterization holds in a much wider regime of parameters.

## 1   Introduction

Characterization of different learning tasks using a combinatorial condition has been investigated in depth in machine learning. Learning a class in an unconstrained fashion is characterized by a finite VC dimension [40, 9], and weakly learning in the statistical query (SQ) framework is characterized by a small SQ dimension [7]. Is there a simple combinatorial condition that characterizes learnability with bounded memory? In this paper we propose a candidate condition, prove upper and lower bounds that match in some of the regime of parameters, and conjecture that they match in a much wider regime of parameters.

A learning algorithm that uses $b$ bits of memory, $m$ samples, and accuracy $1 - \epsilon$ is defined as follows: the algorithm receives a series of $m$ labeled examples one by one, while only preserving an internal state in $\{0, 1\}^b$ between examples. In this paper we focus our attention on the realizable setting: the labeled examples are pairs $(x_i, c(x_i))$, where $x_i \in \mathcal{X}$ and $c : \mathcal{X} \to \{-1, 1\}$ is a concept in a concept class $\mathcal{C}$. The algorithm is supposed to return with constant probability a hypothesis $h$ which matches the unknown concept $c$ on a $1 - \epsilon$ fraction of the underlying distribution. In this paper we further assume that the underlying distribution $P$ on $\mathcal{X}$ is known to the learner, similar to the setting in the SQ framework.

There are two "trivial" algorithms for the problem which we now present. For ease of presentation, we restrict our attention in the introduction to a small constant $\epsilon$, say $\epsilon = 0.01$. Without making any additional assumptions, the following space complexity bounds are known when learning with accuracy 0.99:

1. The ERM algorithm keeps in memory $m = O(\log |\mathcal{C}|)$ samples, and outputs a hypothesis that is consistent with the entire sample. This requires $b = O(\log |\mathcal{C}| \log |\mathcal{X}|)$ bits.

2. A learning algorithm that enumerates all possible concepts in $\mathcal{C}$ and the consistency of each concept based on few random samples. This algorithms requires $m = O(|\mathcal{C}| \log |\mathcal{C}|)$ samples and $b = O(\log |\mathcal{C}|)$ bits.

We define a class $\mathcal{C}$ under a distribution $P$ to be *learnable with bounded memory* if there is a "non-trivial" learning algorithm with respect to both sample complexity and space complexity. A bit more formally, if there is a learning algorithm that uses only $m = |\mathcal{C}|^{o(1)}$ samples and $b = o(\log |\mathcal{C}| \log |\mathcal{X}|)$ bits (see Definition 1).

A crucial combinatorial measure that has been linked to bound-memory *weak* learning is the *statistical query* (SQ) dimension (see Definition 3). Extending these results to *strong* learning requires the following definition. We say that a distribution $Q$ is $\mu$-close to the distribution $P$ (where $\mu \geq 1$) if the ratio $P(x)/Q(x)$ is between $1/\mu$ and $\mu$ for all points $x$ in the domain. We denote by $\mathcal{P}_\mu(P)$ the set of all distributions which are $\mu$-close to $P$ (see Definition 2).

Our main results are upper and lower bounds on bounded memory learning, in terms of the SQ dimension of distributions in the neighbourhood of the underlying distribution $P$. While deriving tighter bounds that hold in a wider regime remains an important open question, these are the first characterizations of the space complexity of *strong* learning using the SQ dimension.

1. Suppose that there is a parameter $d \geq 1$ such that for any distribution $Q \in \mathcal{P}_d(P)$ it holds that $\mathrm{SQ}_Q(\mathcal{C}) \leq d$. Then there exists an algorithm that learns the class $\mathcal{C}$ with accuracy 0.99 under the distribution $P$ using $b = O(\log(d) \cdot \log |\mathcal{C}|)$ bits and $m = \mathrm{poly}(d) \cdot \log(|\mathcal{C}|) \cdot \log \log(|\mathcal{C}|)$ samples.

2. If the class $\mathcal{C}$ is PAC-learnable under $P$ with accuracy 0.99 using $b$ bits and $m$ samples, then for every distribution $Q \in \mathcal{P}_{\Theta(1)}(P)$ its SQ dimension is bounded by $SQ_Q(\mathcal{C}) \leq \max(\mathrm{poly}(m), 2^{O(\sqrt{b})})$.

In Section 1.2 we give a more detailed account of the bounds for general $\epsilon$. We show that for small enough $\epsilon$, the two conditions coincide and we in fact get a characterization of bounded memory learning. We conjecture that the characterization holds for a larger range of parameters (see Conjecture 1). We also prove similar conditions for SQ learning, thus implying equivalence between bounded memory learning and SQ learning for small enough $\epsilon$.

## 1.1 Problem setting

In this paper we consider two learning frameworks: a) The PAC model [39] and b) The Statistical Query framework [21]. See a recap of these frameworks in Appendix A.

**Bounded memory learning.** A bounded memory learning algorithm observes a sequence of labeled examples $(x_1, y_1), (x_2, y_2), \ldots$ in a streaming fashion, where $x_i \in \mathcal{X}, y_i \in \{-1, 1\}$. We assume in this paper that the data is realizable, namely $y_i = c(x_i)$ for some concept $c \in \mathcal{C}$. The algorithm maintains a state $Z_t \in \{0, 1\}^b$ after seeing the first $t$ examples, and updates it after seeing the next example to $Z_{t+1} = \psi_t(Z_t, (x_{t+1}, y_{t+1}))$ using some update function $\psi_t$.[1] The parameter $b$ is called the *bit complexity* of the algorithm. Finally, after observing $m$ samples (where $m$ is a parameter tuned by the algorithm), a hypothesis $h = \phi(Z_m)$ is returned.

We now expand the "trivial" learning algorithms described earlier to accuracy $1 - \epsilon$ for any $\epsilon > 0$:

1. We can learn with accuracy $1 - \epsilon$ using $m = O\left(\log |\mathcal{C}| \mathrm{poly}(1/\epsilon)\right)$ samples and number of bits equal to $b = O\left(\log |\mathcal{C}| \log |\mathcal{X}| + \log |\mathcal{C}| \log(1/\epsilon)\right)$. For constant accuracy parameter this can be done by saving $O(\log |C|)$ examples and applying ERM. To achieve better accuracy we can apply Boosting-By-Majority [15] as we describe in Section 3.

2. One can always learn with $m = O(|\mathcal{C}| \log |\mathcal{C}| \epsilon^{-1})$ samples and $b = O(\log |\mathcal{C}|)$ bits, by going over all possible hypothesis and testing if the current hypothesis is accurate on a few random samples.

We define a class $\mathcal{C}$ to be bounded memory learnable if there is a learning algorithm that beats both of the above learning algorithms. Bounded-memory algorithms should be allowed to save at least a hypothesis and an example in memory. But in extreme cases saving one hypothesis means allowing saving the entire training data in memory. Thus, the definition is most appropriate for the case that $|\mathcal{C}|$ is about the same as $|\mathcal{X}|$.

**Definition 1** (Bounded memory learnable classes). *A class $\mathcal{C}$ under a distribution $P$ is* learnable with bounded memory *with accuracy $1 - \epsilon$ if there is a learning algorithm that uses only $m = (|\mathcal{C}|/\epsilon)^{o(1)}$ samples and $b = o(\log |\mathcal{C}|(\log |\mathcal{X}| + \log(1/\epsilon)))$ bits*[2].

To illustrate this, consider the case where the number of concepts and points are polynomially related, $|\mathcal{C}|, |\mathcal{X}| = \text{poly}(N)$, and where the desired error in not too tiny, $\epsilon \geq 1/\text{poly}(N)$. Then a non-trivial learning algorithm is one that uses a sub-polynomial number of samples $m = N^{o(1)}$ and a sub-quadratic number of bits $b = o(\log^2 N)$. There are classes that can not be learned with bounded memory.

**Example 1** (Learning parities). *Consider the task of learning parities on $n$ bits. Concretely, let $N = 2^n$, $\mathcal{X} = \mathcal{C} = \{0,1\}^n$, $P$ be the uniform distribution over $\mathcal{X}$, and let the label associated with a concept $c \in \mathcal{C}$ and point $x \in \mathcal{X}$ be $\langle c, x \rangle$ (mod 2). It was shown by [29, 26] that achieving constant accuracy for this task requires either $b = \Omega(n^2) = \Omega(\log^2 N)$ bits of memory or an exponential in $n$ many samples, namely $m = 2^{\Omega(n)} = N^{\Omega(1)}$ samples.*

**Close distributions.** An important ingredient in this work is the notion of nearby distributions, where the distance is measured by the multiplicative gap between the probabilities of elements.

**Definition 2** ($\mu$-close distributions). *We say that two distributions $P, Q$ on $\mathcal{X}$ are $\mu$-close for some $\mu \geq 1$ if $\mu^{-1}P(x) \leq Q(x) \leq \mu P(x)$ for all $x \in \mathcal{X}$. Note that the definition is symmetric with respect to $P, Q$. We denote the set of all distributions that are $\mu$-close to $P$ by $\mathcal{P}_\mu(P)$.*

## 1.2 Main results

**Bounded memory PAC learning.** We state our main results for a combinatorial characterization of bounded memory PAC learning in terms of the SQ dimension of distributions close to the underlying distribution.

**Theorem 1.** *Let $\epsilon \in (0,1)$, $d \in \mathbb{N}$ and denote by $\mu = \Theta(\max\{d, 1/\epsilon^3\})$. Suppose that the distribution $P$ satisfies the following condition: for any distribution $Q \in \mathcal{P}_\mu(P)$, $\text{SQ}_Q(\mathcal{C}) \leq d$. Then there exists an algorithm that learns the class $\mathcal{C}$ with accuracy $1 - \epsilon$ under the distribution $P$ using $b = O(\log(d/\epsilon) \cdot \log |\mathcal{C}|)$ bits and $m = \text{poly}(d/\epsilon) \cdot \log(|\mathcal{C}|) \cdot \log\log(|\mathcal{C}|)$ samples.*

**Theorem 2.** *If a class $\mathcal{C}$ is strongly PAC-learnable under $P$ with accuracy $1 - 0.1\epsilon$ using $b$ bits and $m$ samples, then for every distribution $Q \in \mathcal{P}_{1/\epsilon}(P)$, its SQ-dimension is bounded by $SQ_Q(\mathcal{C}) \leq \max\left(\text{poly}(m/\epsilon), 2^{O(\sqrt{b})}\right)$.*

There is a regime of parameters where the upper and lower bounds match. Let $|\mathcal{C}|, |\mathcal{X}| = \text{poly}(N)$ and that $\epsilon = N^{-o(1)}$. Recall that the class is bounded memory learnable if there is a learning algorithm with sample complexity $m = N^{o(1)}$ and space complexity $b = o(\log^2 N)$. Let $\mu, d = N^{o(1)}$. We have the following equivalence, which we conjecture holds for any $\epsilon$:

$$\mathcal{C} \text{ is bounded memory learnable under } P \text{ with accuracy } 1 - \epsilon$$
$$\Updownarrow$$
$$\forall Q \in \mathcal{P}_{poly(1/\epsilon)}(P), \text{SQ}_Q(\mathcal{C}) \leq poly(1/\epsilon).$$

**Conjecture 1.** *For any $\epsilon$, the class $\mathcal{C}$ is bounded memory learnable under distribution $P$ with accuracy $1 - \epsilon \iff \forall Q \in \mathcal{P}_{poly(1/\epsilon)}(P), \text{SQ}_Q(\mathcal{C}) \leq poly(1/\epsilon).$*

**SQ learning.** Next, we give our secondary results for SQ learning, which are very similar to our results for bounded memory learning. Conceptually, it shows that the two notions are tightly connected.

**Theorem 3.** *Let $\epsilon \in (0,1)$, $d \in \mathbb{N}$ and denote by $\mu = \Theta(\max\{d, 1/\epsilon^3\})$. Suppose that the distribution $P$ satisfies the following condition: for any distribution $Q \in \mathcal{P}_\mu(P)$, $\mathrm{SQ}_Q(\mathcal{C}) \leq d$. Then there exists an SQ-learner that learns the class $\mathcal{C}$ with accuracy $1 - \epsilon$ under the distribution $P$ using $q = \mathrm{poly}(d/\epsilon)$ statistical queries with tolerance $\tau \geq \mathrm{poly}(\epsilon/d)$.*

**Theorem 4.** *If a class $\mathcal{C}$ is strongly SQ-learnable under $P$ with accuracy $1 - 0.1\epsilon$, $q$ statistical queries, and tolerance $\tau$, then for every distribution $Q \in \mathcal{P}_{1/\epsilon}(P)$, $SQ_Q(\mathcal{C}) \leq \mathrm{poly}(q/\epsilon\tau)$.*

Note that for any class $\mathcal{C}$, underlying distribution and accuracy $1 - \epsilon$, one can SQ-learn the class with $q = |\mathcal{C}|$ statistical queries and tolerance $\tau = O(\epsilon)$, by going over all the hypotheses. Thus a class is non-trivially SQ-learnable if one can learn it with $q = |C|^{o(1)}$ queries and tolerance $\tau \geq \mathrm{poly}(\epsilon)$. Focusing on the case that $|\mathcal{C}|, |\mathcal{X}| = \mathrm{poly}(N)$ and $\mu, d, q, 1/\epsilon, 1/\tau = N^{o(1)}$, we get that bounded memory learning is equivalent to SQ learning.

## 1.3 Related work

**Characterization of bounded memory learning.**   Many works have proved lower bounds under memory constraints [34, 29, 23, 25, 26, 30, 17, 12, 4, 35, 18, 11]. Some of these works even provide a necessary condition for learnability with bounded memory. As for upper bounds, not many works have tried to give a general property that implies learnability under memory constraints. One work suggested such property [27] but this did not lead to a full characterization of bounded memory learning.

**Statistical query learning.**   After Kearns's introduction of statistical query [21], Blum et al. [7] characterized weak learnability using SQ dimension. Specifically, if $SQ_P(\mathcal{C}) = d$, then $\mathrm{poly}(d)$ queries are both needed and sufficient to learn with accuracy $1/2 + \mathrm{poly}(1/d)$. Note that the advantage is very small, only $\mathrm{poly}(1/d)$. Subsequently several works [3, 36, 38, 13] suggested a few characterizations of strong SQ learnability.

**Bounded memory and SQ dimension.**   In this paper we prove an equivalence, in some parameters regime, between bounded memory learning and SQ learning. There were a few indications in the literature that such an equivalence exists. The work [37] showed a general reduction from any SQ learner to a memory efficient learner. Alas, they gave an example that suggests that an equivalence is incorrect, which we now address.

**Example 2** (Learning sparse parity)**.** *Consider the concept class of parity on the first $k$ bits of an $n$-bit input for $k \ll n$, for example $k = \sqrt{n}$. That is, $\mathcal{X} = \{0,1\}^n$ and $\mathcal{C} = \{0,1\}^k \cdot \{0\}^{n-k}$ is a subset of all possible parities. Naively, an ERM algorithm would need to store $\Theta(k)$ examples, each requiring $n$ bits, and hence need $b = \Theta(kn)$. However, it suffices to store only the first $k$ bits of each example, and hence only use $b = \Theta(k^2)$ bits. As this is significantly less than the naive bound of $\Theta(kn)$ we consider the class to be bounded memory learnable. On the other hand, the SQ dimension of $\mathcal{C}$ is maximal, namely $2^k$, and hence [37] suggest that this example separates bounded memory learning and SQ learning.*

*Relating to our results, it shows two things: when the sizes of the concept classes $\mathcal{C}$ and example set $\mathcal{X}$ are polynomially related, there is no such separation (we prove this for small enough $\epsilon$ and conjecture for all $\epsilon$). Moreover, the $2^{O(\sqrt{b})}$ term in Theorem 2 is tight.*

The work [17] showed that high SQ dimension implies non-learnability with bounded memory when the learner returns the exact answer. However, learnability is usually inexact and this does not relate to strong learnability.

**Littlestone dimension.**   Online learnability without memory constraints is characterized using Littlestone dimension [24]. This dimension is not suited for bounded memory learning as it does not take into account the structure of the class which determines whether the class is learnable with bounded memory or not. Specifically, there are problems that have similar Littlestone dimension (e.g., parity and discrete thresholds on the line), where the former (thresholds) is easy to learn under memory constraints and the latter (parity) is hard.

**Learning under a known distribution.**   In SQ framework most works focused on learning under known distributions [7, 10, 43, 44, 3, 36, 13, 38]. However, PAC learning research under known

distribution is scarce but exists, e.g., [6, 5, 41, 31]. In particular, Benedek et al. [6] showed that unconstrained learning under known distribution is characterized by covering.

**Smooth distributions.**  A key idea in this paper is to use *close* distributions which are upper and lower bounded by a distribution. A one sided closeness, namely the upper bound, is referred in the literature as a *smooth* distribution, see for example [10]. Smooth distributions were also used to show equivalence between boosting and hard-core sets [22, 19].

**Paper organization.**  We begin in Section 2 with a presentation of known results in boosting and statistical queries that we will need. In Section 3 we construct learning algorithms based on the assumption that close distributions have bounded SQ dimensions, and prove Theorem 1 and Theorem 3. In Section 4 we establish the reverse direction and prove Theorem 2 and Theorem 4. Omitted proofs can be found in the appendix.

## 2   Preliminaries

**Weak learning and boosting.**  It is often conceptually easier to design an algorithm whose accuracy is slightly better than an educated guess, and then attempt to boost its accuracy.

Consider first the PAC model. We say that a learning algorithm $\mathcal{W}$ is a $\gamma$-*weak learner* if there exists an integer $m$ such that for any target concept $c \in \mathcal{C}$ and any $n \geq m$, with probability at least $2/3$ over the draw of an i.i.d. labeled sample $S = ((x_1, c(x_1)), \ldots, (x_n, c(x_n)))$ according to the underlying distribution $P$, the hypothesis returned by $\mathcal{A}$ is $(1/2 - \gamma)$-accurate. We refer to the minimal integer $m$ satisfying the above as the *sample complexity* of the weak learner. The notion of $\gamma$-weak learning in the SQ framework is defined analogously, where the *query complexity* of the weak learner is denoted by $q_\tau$ (where $\tau$ is the tolerance parameter).

A *boosting* algorithm $\mathcal{A}$ uses an oracle access to a weak learner $\mathcal{W}$ and aggregates the predictions of $\mathcal{W}$ into a satisfactory accurate solution. The celebrated works of Freund and Schapire [32, 33, 14, 15, 16] provide several successful boosting algorithms for the PAC model. The work of [2] extended some of these results to the SQ framework.

**Known SQ-dimension bounds for weak learning.**  The following upper and lower bounds are known. The first upper bound is a folklore lemma whose proof can be found in [38].

**Proposition 1.** *Let $\mathcal{C}$ be a concept class, $P$ an underlying distribution, such that $\mathrm{SQ}_P(\mathcal{C}) \leq d$. Then there is a $(1/d)$-weak SQ-learner with query complexity $q = d$ and tolerance $\tau = 1/3d$.*

The next lower bound was initially proved by [8]. A simplified proof was given later by [38].

**Proposition 2.** *Let $\mathcal{C}$ be a concept class, $P$ an underlying distribution, and let $d = SQ_P(\mathcal{C})$. Any learning algorithm that uses tolerance parameter lower bounded by $\tau > 0$ requires in the worst case at least $(d\tau^2 - 1)/2$ queries for learning $\mathcal{C}$ with accuracy at least $1/2 + 1/d$.*

Finally, the next proposition shows that SQ learnability (weak or strong) implies learning with bounded memory.

**Proposition 3** (Theorem 7 in [37])**.** *Assume that a class $\mathcal{C}$ can be learned using $q$ statistical queries with tolerance $\tau$. Then there is an algorithm that learns $\mathcal{C}$ using $m = O(\frac{q \log |\mathcal{C}|}{\tau^2}(\log(q) + \log\log(|\mathcal{C}|))$ samples and $b = O(\log |\mathcal{C}| \cdot \log(q/\tau))$ bits.*

## 3   From bounded SQ dimension to bounded memory learning

In this section we prove our upper bounds: Theorem 1 and Theorem 3. A schematic illustration of the proof is given in Fig. 1.

**Overview.**  To prove Theorem 1 we apply an extension of the Boosting-By-Majority (BBM) algorithm [15] to the SQ framework due to [2]. Similarly to other popular boosting methods (e.g. AdaBoost [16]), the algorithm operates by re-weighting the input sample and feeding the weak learner with sub-samples drawn according to the re-weighed distributions. The main challenge is to bound

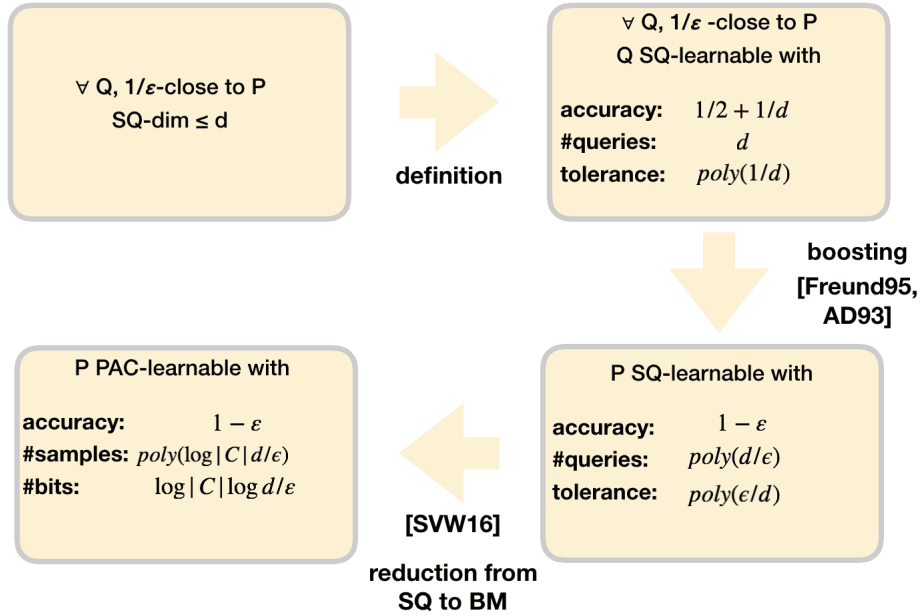

Figure 1: Proof outline (with asymptotic terms): from bounded SQ dimension under close distributions to strong learnability.

the SQ-dimension of the probability distributions maintained by the boosting algorithm. This will allow us to obtain a bound on the query complexity of the boosting process using Proposition 1 and thus conclude Theorem 1. Consequently, we deduce Theorem 3 using Proposition 3.

**SQ-Boost-By-Majority.** Following [2] we describe how BBM can be carried out in the SQ model. Instead of having an access to a sampling oracle, the booster $\mathcal{A}$ has an access to an SQ oracle with respect to the distribution $P$ and the target concept $c$. Similarly to BBM, the booster re-weights the points in $\mathcal{X}$ in iterative fashion, thereby defining a sequence of distributions, $P_1, \ldots, P_T$. The weak learner $\mathcal{W}$ itself also works in the SQ model. That is, instead of requiring samples $S_1, \ldots, S_T$ drawn according to $P_1, \ldots, P_T$, it submits statistical queries to the boosting algorithm. The guarantee of the weak learner remains intact; provided that it gets sufficiently accurate answers (as determined by the tolerance parameter $\tau$), $\mathcal{W}$ should output a weak classifier whose correlation with the target concept is at least $\gamma$.

Therefore, the challenging part in translating BBM to the SQ model is to enable simulating answers to statistical queries with respect to the distributions $P_1, \ldots, P_T$ given only an access to an SQ oracle with respect to the initial distribution $P$. Fortunately, the BBM's re-weighting scheme makes it rather easy. It follows from the definition of the distributions maintained by BBM (see Eq. (1) and Eq. (2)) that in the beginning of round $t$, the space $\mathcal{X}$ partitions into $t$ regions such that the probability of points in each region is proportional to their initial distribution according to $P$. This allows simulating an *exact* SQ query with respect to $P_{t+1}$ using $O(t)$ exact SQ queries to $P$. Furthermore, as shown in [2], the fact that $P_t(x) \leq (C/\epsilon^3) \cdot P(x)$ allows us to perform this simulation with suitable tolerance parameters. This is summarized in the next proposition.

**Proposition 4** ([2]). *Any statistical query with respect to the distribution $P_t$ with tolerance $\tau$ can be simulated using $O(t)$ statistical queries with respect to the original distribution $P$ with tolerance parameter $\Omega(\tau \cdot \mathrm{poly}(\epsilon))$.*

**Upper bounding the SQ-dimension of SQ-BBM's distributions.** In this part we derive an upper bound on the SQ-dimension of the distribution $P_1, \ldots, P_T$ maintained by SQ-BBM. To this end we use our assumption that for all $Q \in \mathcal{P}_\mu(P)$, $\mathrm{SQ}_Q(\mathcal{C}) \leq d$ where $\mu = \max\{C/\epsilon^3, 4d\}$. While we

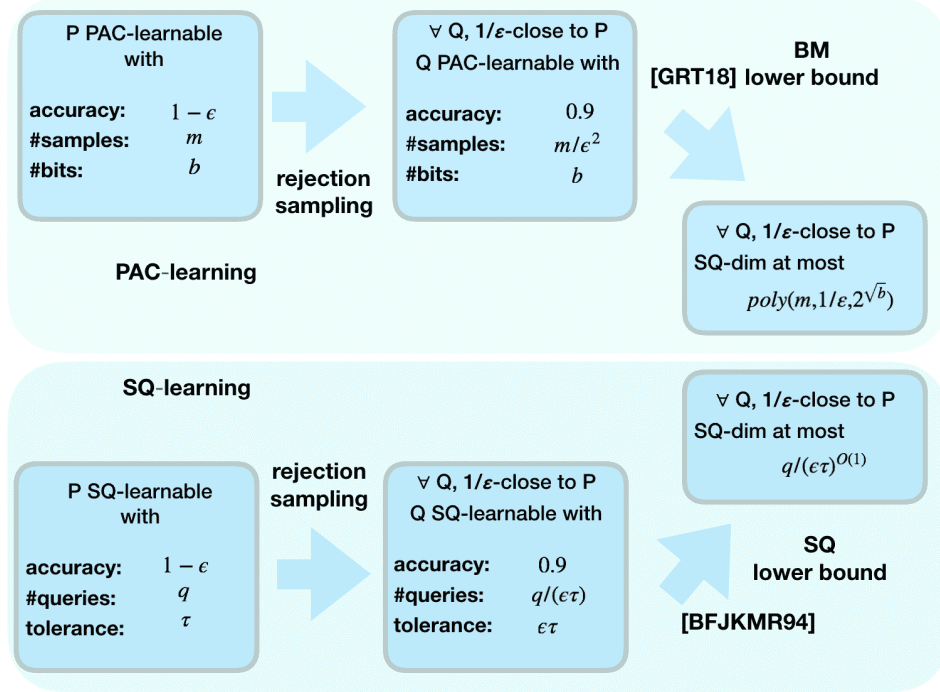

Figure 2: Proof outline (with asymptotic terms): from strong learnability to bounded SQ-dimension under close distributions in PAC and SQ models.

cannot make sure that the distributions $P_1, \ldots, P_T$ belong to $\mathcal{P}_\mu(P)$, we will still be able to derive an upper bound on their SQ-dimension.

**Lemma 1.** *Let $P_1, \ldots, P_T$ be the distributions maintained by SQ-BBM. For every $t = 1, \ldots T$, $\mathrm{SQ}_{P_t}(\mathcal{C}) \le 4d$.*

**Putting it all together.**    We now complete the proofs of Theorem 3 and Theorem 1.

*Proof of Theorem 3.* From Proposition 1 we conclude that for any $Q \in \mathcal{P}_\mu(P)$ there exists a $(1/d)$-weak learner with query complexity $d$ and tolerance $1/(3d)$. Using this weak learner we apply SQ-BBM as described above. From Lemma 1 we know that for every distribution $P_t$ maintained by SQ-BBM, $\mathrm{SQ}_{P_t}(\mathcal{C}) = O(d)$. Combining Proposition 6 and Proposition 4 we conclude that SQ-BBM reaches a $1 - \epsilon$ accurate prediction after $T = O(\mathrm{poly}(d) \log(1/\epsilon))$ iterations while using at most $\mathrm{poly}(d/\epsilon)$ statistical queries with tolerance at least $\mathrm{poly}(\epsilon/d)$. □

*Proof of Theorem 1.* Proposition 3 tells us that if a class $\mathcal{C}$ can be learned using $q$ statistical queries with tolerance $\tau$, then there is a PAC algorithm that learns $\mathcal{C}$ using $m = O(\frac{q \log |\mathcal{C}|}{\tau^2}(\log(q) + \log \log(|\mathcal{C}|))$ samples and $b = O(\log |\mathcal{C}| \cdot \log(\frac{q}{\tau}))$ bits. Theorem 3 gives an SQ learning algorithm $q = \mathrm{poly}(d/\epsilon)$ and $\tau \ge \mathrm{poly}(\epsilon/d)$, which gives a bounded memory learning algorithm with $m = \mathrm{poly}(d/\epsilon) \cdot \log |\mathcal{C}| \cdot \log \log |\mathcal{C}|$ samples and $b = O(\log |\mathcal{C}| \cdot \log(d/\epsilon))$ bits.

□

## 4    From bounded memory learning to bounded SQ dimension

In this section we prove our lower bounds: Theorem 2 and Theorem 4. A schematic illustration of the proof is given in Fig. 2.

**Overview.**    We use the rejection sampling technique to transform a given strong learner with respect to distribution $P$ into a weak learner with respect to any close distribution $Q$. This can be established

both in the PAC learning framework and the SQ framework. By virtue of Proposition 2, this implies Theorem 4. To prove Theorem 2, we would like to use a recent result by [17] that establishes an upper bound on $SQ_Q(\mathcal{C})$ given memory-efficient learner. Unfortunately, the derivation in [17] requires the learner to return the *exact* target concept. Our weak learner does not necessarily satisfy this requirement. In fact, it is even not necessarily proper, i.e., it might return a hypothesis $h \notin \mathcal{C}$. To get around this obstacle, we first show how to transform any improper weak learning rule into a proper learning rule. Then, we focus on the hypotheses $\mathcal{H} \subseteq \mathcal{C}$ that constituents that SQ dimension, i.e, $SQ_Q(\mathcal{H}) = SQ_Q(\mathcal{C})$. We ensure that the exact target concept $c$ is returned, as large $SQ_Q(\mathcal{H})$ implies that all hypotheses in $\mathcal{H}$ are far a part.

**From strong learning to weak learning of close distributions.** The next claim shows that if a class is strongly learnable under distribution $P$, then it is weakly learnable under *any* close distribution $Q$. The idea is to utilize the closeness assumption in order to perform rejection sampling from $Q$ to simulate sampling from $P$.

**Lemma 2.** *Let $P$ be a distribution over $\mathcal{X}$. Assume that the concept class $\mathcal{C}$ can be learned with accuracy $1 - 0.1\epsilon$, $m$ samples, and $b$ bits under distribution $P$. Then, any probability distribution $Q$ that is $(1/\epsilon)$-close to $P$ can be learned with accuracy $0.9$, $O(m/\epsilon^2)$ samples, and $b$ bits.*

**Rejection sampling algorithm in the SQ model.** Analogously to Lemma 2, we can show that also under the SQ framework, strong learning implies weak learning of close distributions. The proof uses the same rejection sampling technique as in Lemma 2.

**Lemma 3.** *Let $P$ be a distribution over $\mathcal{X}$. Assume the concept class $\mathcal{C}$ can be learned with accuracy $1 - 0.1\epsilon$, $q$ queries and tolerance $\tau$ under distribution $P$. Then, any probability distribution $Q$ that is $(1/\epsilon)$-close to $P$ can be SQ-learned with accuracy $0.9$ using $O(q/\epsilon\tau)$ queries with tolerance $\epsilon\tau/2$.*

**From weak learning to low SQ-dimension.** The next few claims establish the fact that if a class $\mathcal{C}$ is learnable with bounded memory under distribution $Q$, the statistical dimension $SQ_Q(\mathcal{C})$ is low.

**Proposition 5** (Corollary 8 in [17]). *Let $\mathcal{H} = \{h_1, \ldots, h_d\}$ a class and $Q$ a distribution with $SQ_Q(\mathcal{H}) = d$. Any learning algorithm that uses $m$ samples, $b$ bits and returns the exact correct hypothesis with probability at least $\Omega(1/m)$ must use at least $m = d^{\Omega(1)}$ samples or $\Omega(\log^2 d)$ bits.*[3]

The algorithm described in the previous section will not return the exact hypothesis, and more generally will not even be a proper learner (i.e., it will not necessarily return a hypothesis from the class). Fortunately, we can transform any improper learner into a proper learner without significantly increasing the neither the sample nor the space complexity.

**Lemma 4.** *Fix a class $\mathcal{C}$. Let $\mathcal{A}$ be an improper learning algorithm for $\mathcal{C}$ that uses $b$ bits, $m$ samples, and accuracy $1 - \epsilon$. Then there is an $(1 - 3\epsilon)$-accurate proper learning algorithm that uses $O(m)$ samples and $b + O(\log(|\mathcal{C}|/\epsilon))$ bits.*

**Lemma 5.** *Fix a class $\mathcal{C}$ and a distribution $Q$. If $\mathcal{C}$ is learnable with accuracy $0.9$ under $Q$ using $m$ samples and $b$ bits, then*

$$SQ_Q(\mathcal{C}) \leq \max(m^{O(1)}, 2^{O(\sqrt{b})}).$$

**Putting it all together.** We now complete the proofs of Theorem 2 and Theorem 4.

*Proof of Theorem 2.* Assume the concept class $\mathcal{C}$ can be learned with accuracy $1 - 0.1\epsilon$, $m$ samples, and $b$ bits under distribution $P$. Lemma 2 states that any distribution $Q$ that is $(1/\epsilon)$-close to $P$ can be learned with accuracy $0.9$, $O(m/\epsilon^2)$ samples, and $b$ bits. Lemma 5 completes the claim. □

*Proof of Theorem 4.* Assume that the concept class $\mathcal{C}$ can be learned with accuracy $1 - 0.1\epsilon$, $q$ queries and tolerance $\tau$ under distribution $P$. Lemma 3 states that any distribution $Q \in \mathcal{P}_{1/\epsilon}(P)$ can be SQ-learned with accuracy $0.9$, $O(m/\epsilon\tau)$ queries, and tolerance $\epsilon\tau/2$. Proposition 2 completes the claim. □

## Broader Impact

Algorithms with bounded memory are extensively studied ([1],[42],[28]). But bounded memory *learning* algorithms were only recently been investigated. In machine learning we have a good understanding of PAC learning using the VC dimension; weak learning with statistical queries using the SQ dimension; and online learning using the Littlestone dimension. An understanding of bounded-memory learning is missing. There are many works showing lower bounds, but none that shows both upper and lower bounds.

We are the first to (1) give a characterization of bounded-memory learning in some regime, (2) in this regime we show equivalence to a different and known framework, statistical queries.

Our impact is two-fold: for the general ML community we give an understanding of the capabilities and limitations of bounded-memory learning and we show its equivalence to a known framework. Second, for the theory researchers, we leave many open problems:

1. Proving a characterization for the entire regime.
2. Utilizing the equivalence between statistical queries and bounded memory to gain a better understanding of these two frameworks.
3. Our work focused on the case that $|C|, |X|$ are polynomially related. We leave for future research to investigate the regimes of $|C| = |X|^{o(1)}$ and $|X| = |C|^{o(1)}$.

## Acknowledgments and Disclosure of Funding

This work is supported by NSF award 1909634

## Footnotes

[1]Following the model of branching programs (e.g., [29]), the maps $\psi_1, \psi_2, \ldots$ are not considered towards the space complexity of the algorithm.

[2]Formally, the $o(\cdot)$ factors are in terms of the size of the class $\mathcal{C}$. Hence this definition applies to families of distributions $\{\mathcal{C}_n\}$ of growing size, for example parities on $n$ bits. However, in the main theorems we give quantitative bounds and hence can focus on single classes instead of families of classes.

[3]In [17] they consider the case where $Q$ is the uniform distribution. By creating a few copies of the examples in $\mathcal{X}$ we can transform a general *known* distribution to be as close as to uniform as needed. Note that the size of the domain $\mathcal{X}$ is not a relevant parameter here.

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
