[Supplementary Material]

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

[4]Given a confidence parameter $\delta > 2/3$, standard amplification techniques can be used to ensure that the probability error is at most $\delta$, while increasing the sample complexity by at most a $\log(1/\delta)$ multiplicative factor.

[5]According to the original framework of Kearns, (seemingly) more general queries are allowed. Namely, each query is a pair $(\chi, \tau)$ where $\chi : \mathcal{X} \times \{-1, 1\} \to \{-1, 1\}$. The oracle has to answer the query with a scalar $\nu$ satisfying

[7]In [15], the algorithm does not actually abort but proceeds by drawing random hypotheses for $T - t$ rounds. It was shown in [20], Lemma 5.2, that (with the above rejection criteria) the algorithm can actually abort and return a majority vote.

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

# A  Background

**PAC model.**    In PAC learning [39] we consider the task of binary classification over an *instance space* $\mathcal{X}$. Denote by $\mathcal{C} \subseteq \{-1, 1\}^{\mathcal{X}}$ a concept class of functions mapping instances to binary labels, and let $c \in \mathcal{C}$ be the *target* (a.k.a. true) concept. Also, let $P$ be the underlying probability distribution over $\mathcal{X}$. We assume that $P$ is known to the learner whereas the target concept $c$ is not known.

The input to the learning algorithm $\mathcal{A}$ consists of a labeled sample $S = ((x_1, c(x_1)), \ldots, (x_m, c(x_m)))$ such that $S_X := (x_1, \ldots, x_m) \sim P^m$. Its output has the form of a hypothesis $h \in \{-1, 1\}^{\mathcal{X}}$. We measure the success of the algorithm according to its expected error $L_{P,c}(h) = \Pr_{x \sim P}(h(x) \neq c(x))$. We say that $h$ is $\epsilon$-accurate if $L_{P,c}(h) \leq \epsilon$. The *sample complexity* of $\mathcal{A}$ under the distribution $P$, denoted $m(\epsilon) : (0, 1) \to \mathbb{N}$, is a function mapping a desired accuracy $\epsilon$ to the minimal positive integer $m(\epsilon)$ such that for any target concept $c \in \mathcal{C}$ and any $m \geq m(\epsilon)$, with probability at least $2/3$ over the drawn of an i.i.d. sample $S = ((x_1, c(x_1)), \ldots, (x_m, c(x_m)))$, the output $\mathcal{A}(S)$ is $\epsilon$-accurate.[4]

**The statistical query framework.**    The statistical query (SQ) framework has been introduced by [21] to handle random noise in the PAC setting. In this model, instead of having access to an i.i.d. sequence of labeled instances, the learner has access to a *statistical query oracle* (a.k.a. *correlation oracle*). Each call to the oracle has the form of a pair $(h, \tau)$, where $h \in \{-1, 1\}^{\mathcal{X}}$ is a hypothesis and $\tau > 0$ is called a *tolerance* parameter. The oracle has to answer such a query with a scalar $\nu$ satisfying[5]

$$|\langle h, c \rangle_P - \nu| \leq \tau \quad \text{where} \quad \langle h, c \rangle_P := \mathbb{E}_{x \sim P}[h(x)c(x)].$$

As was shown in [21], any approximately accurate algorithm in the SQ model can be efficiently transformed into an approximately accurate PAC algorithm, i.e. an algorithm that has access to i.i.d. labeled examples. The resulted PAC is also robust to noise. We refer to [38] for additional background.

Analogously to the definition of sample complexity, the *query complexity* of a learning algorithm in the SQ model, denoted $q_\tau(\epsilon)$, is the minimal number of queries with tolerance parameter $\tau$ required for achieving $\epsilon$-accurate prediction (for any target concept $c \in \mathcal{C}$).

**SQ dimension.**    The SQ-dimension defined below is useful for characterizing weak learnability in the statistical query framework, as was proved in [7] (see Proposition 1 and Proposition 2).

**Definition 3** (Statistical query dimension). *Fix a probability distribution $P$ over $\mathcal{X}$. The SQ-dimension of the class $\mathcal{C}$ with respect to the distribution $P$, denoted $\mathrm{SQ}_P(\mathcal{C})$, is the maximal integer $d$ such that there exist $h_1, \ldots, h_d \in \mathcal{C}$ satisfying $|\langle h_i, h_j \rangle_P| \leq 1/d$ for all $i \neq j \in [d]$.*

**Additional notation**    We denote the density and the cumulative binomial distribution by $\mathrm{Binom}(m, r, p)$ and $\mathrm{Binom}(m, \leq r, p)$, which respectively refer to the probability of observing exactly (at most) $r$ heads in $m$ independent and identical trials where the probability of "head" in each single trial is $p$.[6]

$$|\mathbb{E}_{x \sim P}[\chi(x, c(x))] - \nu| \leq \tau .$$

Note that $\chi(x, c(x))$ can be written as a polynomial in $x$ and $c(x)$, and since $c(x)$ is either $1$ or $-1$, this polynomial is linear in $c(x)$. In other words, $\chi(x, c(x)) = g_1(x)c(x) + g_2(x)$. given that the distribution $P$ is known, $\mathbb{E}_{x \sim P}[g_2(x)]$ can be calculated. Thus, one can simulate the seemingly more general query $\chi$ using the correlation query applied to $g_1$.

[6]If $r > m$ or $r < 0$ then both terms are equal to zero.

## B Omitted Proofs

### B.1 From bounded SQ dimension to bounded memory learning

**Reviewing Boost-By-Majority (BBM).** Let $\mathcal{W}$ be a $\gamma$-weak learner with respect to the distribution $P$ with sample complexity $m_0$. Similarly to most boosting algorithms, BBM operates by iteratively re-weighting and feeding a given $\gamma$-weak learner with $T$ i.i.d. samples $S_1, \ldots, S_T$ of size $m_0$. The outputs $h_1, \ldots, h_T$ of the weak learner are then aggregated into a majority vote classifier:

$$h(x) = \texttt{Majority}(h_1(x), \ldots, h_T(x)) := \begin{cases} 1 & \sum_t h_t(x) > 0 \\ -1 & \text{otherwise} \end{cases}.$$

To make the algorithm memory-efficient [15] suggests to implement the re-weighting using *rejection sampling*. Let $h_1, \ldots, h_t$ be the weak classifiers collected during the first $t$ rounds. At the beginning of round $t + 1$, the algorithm draws an example $x \sim P$ and keeps it with probability

$$w_{t+1}(x) = \text{Binom}\left(T - t, \left\lfloor \frac{T - t - r(x)}{2} \right\rfloor, 1/2 + \gamma\right) \qquad \text{where } r(x) := \sum_{i=1}^{t} h_i(x). \quad (1)$$

Therefore, the induced probability distribution on time $t$ is

$$P_{t+1}(x) = w_{t+1}(x)P(x)/Z \qquad (2)$$

where $Z$ is a normalization factor. It repeats this step until either collecting $m_0$ samples or rejecting $\Theta(\epsilon^{-3} \log T)$ consecutive examples. In the former scenario it feeds the weak learner with the resulted sample, whereas in the latter scenario it aborts the boosting process and returns the hypothesis $h = \texttt{Majority}(h_1(x), \ldots, h_t(x))$.[7]

**Proposition 6.** *[15] Let $\epsilon > 0$. With probability at least 2/3, the following hold:*

1. *BBM reaches an $\epsilon$-accurate hypothesis after at most $T = O(\gamma^{-2} \log(1/\epsilon))$ rounds.*

2. *There exists a global constant $C > 0$ such that for every round $t$, the probability distribution $P_t$ satisfies $P_t(x) \leq (C/\epsilon^3) \cdot P(x)$ for all $x$.*

*Proof of Lemma 1.* Let $\delta = 1/\mu$. Consider the mixed distribution $\tilde{P}_t = \delta P + (1-\delta)P_t$. Proposition 6 implies that for all $x$, $P_t(x) \leq \mu P(x)$. It follows that

$$(\forall x) \qquad \tilde{P}_t(x) \leq \delta P(x) + (1 - \delta)\mu P(x) \leq \mu P(x).$$

Also, clearly we have that

$$(\forall x) \qquad \tilde{P}_t(x) \geq \delta P(x) = \mu^{-1} P(x).$$

Hence, $\tilde{P}_t \in \mathcal{P}_\mu(P)$, and by our assumption we have $\text{SQ}_{\tilde{P}_t}(\mathcal{C}) \leq d$.

Assume by contradiction that there exist $m \geq 4d$ hypotheses $h_1, \ldots, h_m \in \mathcal{C}$ such that

$$|\langle h_i, h_j \rangle_{P_t}| \leq 1/m \qquad (\forall i \neq j \in [m]).$$

Therefore, for all $i \neq j \in [m]$,

$$\left|\langle h_i, h_j \rangle_{\tilde{P}_t}\right| = |\delta\langle h_i, h_j \rangle_P + (1 - \delta)\langle h_i, h_j \rangle_{P_t}| \leq \delta + (1 - \delta)\frac{1}{m} \leq \frac{1}{4d} + \frac{1}{4d} = \frac{1}{2d}.$$

In particular, it follows that $|\langle h_i, h_j \rangle_{\tilde{P}_t}| \leq \frac{1}{2d}$ for all $i \neq j \in [2d]$. This contradicts the fact that $\text{SQ}_{\tilde{P}_t}(\mathcal{C}) \leq d$. $\qquad\square$

## B.2    From bounded memory learning to bounded SQ dimension

*Proof of Lemma 2.* Fix a distribution $P$, a class $\mathcal{C}$ and assume that there is an algorithm $\mathcal{A}$ that learns $\mathcal{C}$ under $P$ with accuracy $1 - 0.1\epsilon$, $m$ samples, and $b$ bits. We want to show that for any $(1/\epsilon)$-close distribution $Q \in \mathcal{P}_{1/\epsilon}(P)$ there is an algorithm that learns the class $\mathcal{C}$ under distribution $Q$ with accuracy $0.9$, $O(m/\epsilon^2)$ samples, and $b$ bits.

At a high level, our analysis involves two steps. First, given a close distribution $Q$ we apply the rejection sampling technique to simulate sampling from the original distribution $P$. This enables us to run the algorithm $\mathcal{A}$. Then we translate the accuracy guarantee of $\mathcal{A}$ with respect to $P$ into a an accuracy guarantee with respect to $Q$.

**Rejection sampling.**    In Algorithm 1 we detail the rejection sampling step mentioned above.

---
**Algorithm 1** Learning from examples distributed by $Q$
---
1: Get a labeled example $x$ from $Q$.
2: Accept $x$ with probability $\frac{P(x)}{Q(x)}\epsilon$.
3: Call algorithm $\mathcal{A}$ with the accepted examples.
---

We first note that the rejection sampling is well defined. Namely, by the closeness assumption, $\frac{P(x)}{Q(x)}\epsilon \in [0,1]$. The distribution induced by the rejection sampling is proportional to $P$ since

$$Q(x) \cdot \frac{P(x)}{Q(x)}\epsilon = P(x)\epsilon.$$

**Strong learning with respect to $P$ $\Rightarrow$ weak learning with respect to $Q$.**    By our assumption on $\mathcal{A}$, with probability at least $2/3$, it outputs a hypothesis $h$ with accuracy at least $1 - 0.1\epsilon$. We next prove that $h$ forms a weak classifier with respect to $Q$. Denoting the target hypothesis by $c \in \mathcal{C}$, we have that

$$L_{Q,c}(h) = \sum_{x:h(x)\neq c(x)} Q(x) \leq \sum_{x:h(x)\neq c(x)} \frac{1}{\epsilon} \cdot P(x) = \frac{1}{\epsilon} \cdot L_{P,c}(h) \leq \frac{1}{\epsilon} \cdot 0.1\epsilon = 0.1 .$$

Thus, the accuracy is at least $0.9$.

So far we proved that the we indeed designed a learning algorithm for $Q$. Let's analyze the parameters of the algorithm. The rejection sampling technique does not require additional bits, thus number of bits is the same as number of bits used in $\mathcal{A}$. We next bound the number of samples needed.

We first note that the probability to accept an example $x$ is $\frac{P(x)}{Q(x)}\epsilon \geq \epsilon^2$, as $Q$ is $(1/\epsilon)$-close to $P$. From Hoeffding's inequality, we know that if we get at least $2m/\epsilon^2$ samples, then the probability that the algorithm does not accept at least $m$ samples is smaller than $e^{-m}$. Thus, with probability at least $1 - me^{-m}$, the number of samples used by the new algorithm is $O(m/\epsilon^2)$.

The confidence of the algorithm is at least $2/3 - e^{-m} \cdot m \geq 7/12$ for large enough $m$. Standard amplification techniques can be used to ensure that the probability error is at most $2/3$, while increasing the sample complexity by at most a constant multiplicative factor.    $\square$

*Proof of Lemma 3.* Fix a distribution $P$, a class $\mathcal{C}$ and assume that there is an algorithm $\mathcal{A}$ that learns $\mathcal{C}$ under $P$ with accuracy $1 - 0.1\epsilon$, $m$ queries, and tolerance $\tau$. Denote the correct hypothesis by $c \in \mathcal{C}$. We want to show that for any $(1/\epsilon)$-close distribution $Q \in \mathcal{P}_{1/\epsilon}(P)$ there is an algorithm that weakly learns the class $C$ under distribution $Q$ in the SQ framework.

Fix a query $\psi$ that is used by $\mathcal{A}$. Ideally, we would like to replace it with a query $\psi'$ of the form

$$\psi'(x) = \begin{cases} \frac{P(x)}{Q(x)}\psi(x) & \text{if } Q(x) \neq 0 \\ 0 & otherwise \end{cases} ,$$

since querying $\psi$ under $P$ is the same as querying $\psi'$ under $Q$, as $\mathbb{E}_Q[\psi'(x)c(x)] = \mathbb{E}_P[\psi(x)c(x)]$. The problem is that the range of $\psi'$ is not $\{-1, 1\}$. To fix it, we will replace $\psi$ with several queries

$\psi_1, \ldots, \psi_n$ that their range is $\{-1, 1\}$ and their average, $\frac{1}{n} \sum_{i=1}^{n} \psi_i$, approximately returns the correct query, i.e., $\psi' \approx \frac{1}{n} \sum_{i=1}^{n} \psi_i$.

For every $x \in \mathcal{X}$ we would like to use Lemma 6 below in order to define $\psi_i(x)$. The first step will be to make sure that $\psi'(x)$ is in $[-1, 1]$. To achieve that we focus on $\epsilon \psi'(x)$, because it is equal to $\epsilon \frac{P(x)}{Q(x)} \psi(x)$ and

$$0 < \epsilon \cdot \frac{P(x)}{Q(x)} \leq \epsilon \cdot \frac{1}{\epsilon} = 1.$$

Using Lemma 6, there are $n = O(1/\epsilon\tau)$ queries $\psi_i$ such that for every $x \in \mathcal{X}$ it holds that

$$\left| \frac{1}{n} \sum_{i=1}^{n} \psi_i(x) - \epsilon \psi'(x) \right| \leq \frac{\epsilon\tau}{2}.$$

From this we can deduce that

$$\left| \frac{1}{\epsilon} \cdot \frac{1}{n} \sum_{i=1}^{n} \mathbb{E}_Q[\psi_i(x)c(x)] - \mathbb{E}_P[\psi(x)c(x)] \right| \leq \frac{\tau}{2}.$$

To summarize, the new learning algorithm $\mathcal{A}'$ that learns under distribution $Q$ will simulate algorithm $\mathcal{A}$ and whenever a query $\psi$ will be needed, it will take $O(1/\epsilon\tau)$ queries created by Lemma 6 and return their average times $1/\epsilon$. Thus, $\mathcal{A}$ uses $O(m/\epsilon\tau)$ queries and its tolerance is $\epsilon\tau/2$.

$\square$

*Proof of Lemma 4.* Fix a class $\mathcal{C}$ and an improper learning algorithm $\mathcal{A}$ for $\mathcal{C}$. Denote the number of bits it uses by $b$, the number of samples by $m$, and the accuracy by $1 - \epsilon$. Define the algorithm $\mathcal{A}'$ as follows:

1. Run algorithm $\mathcal{A}$ that outputs hypothesis $h$ as its answer.

2. Go over all hypothesis in $\mathcal{C}$ and return one that agrees with $h$ on $1 - 2\epsilon$ of the examples by testing consistency on $O(\log |\mathcal{C}|/\epsilon^2)$ random examples.

Note that the second step does not use new samples and requires only $\log |\mathcal{C}| + O(\log(\log |\mathcal{C}|/\epsilon)) = O(\log(|\mathcal{C}|/\epsilon))$ additional bits. The algorithm $\mathcal{A}'$ functions correctly, because by the definition of the algorithm $\mathcal{A}$ there must be hypothesis in $\mathcal{C}$ that agrees on $(1 - \epsilon)$ of the examples. By Hoeffding's inequality, the probability that there is a hypothesis that deviates by more than $\epsilon$ in approximating its loss is small and standard amplification techniques can be used to ensure that the probability error is at most $2/3$, while increasing the sample complexity by at most a constant multiplicative factor. The accuracy of $\mathcal{A}'$ is at least $1 - 3\epsilon$.

$\square$

*Proof of Lemma 5.* Fix a class $\mathcal{C}$ and a distribution $Q$. Assume $\mathcal{C}$ is learnable under $Q$ with $m$ samples, $b$ bits, and accuracy $0.9$. Assume also that $SQ_Q(\mathcal{C}) = d$. Thus, there are $d$ hypotheses $\mathcal{H} = \{h_1, \ldots, h_d\}$ such that $|\langle h_i, h_j \rangle_Q| \leq 1/d$. Since $\mathcal{H} \subseteq \mathcal{C}$ and by our assumption on the learnability of $\mathcal{C}$, we get that $\mathcal{H}$ is learnable under $Q$ with $m$ samples, $b$ bits, and accuracy $0.9$. From Lemma 4, we get that $\mathcal{H}$ is *properly* learnable under $Q$ with $O(m)$ samples, $b + O(\log |\mathcal{H}|)$ bits, and accuracy $0.7$.

We can deduce that there is a learning algorithm for $\mathcal{H}$ that returns the exact hypothesis, as the hypotheses in $\mathcal{H}$ are far apart from each other. Specifically, we know that between any two hypotheses $i \neq j$ there is at least $\frac{1}{2} - \frac{1}{2d}$ disagreement. If $\frac{1}{2} - \frac{1}{2d} > 0.3$, then learning exactly is equivalent to properly learning up to accuracy $0.7$. The equation $\frac{1}{2} - \frac{1}{2d} > 0.3$ is equivalent to $d = \Omega(1)$.

Since the hypotheses in $\mathcal{H}$ are far apart from each other, the number of bits $\mathcal{A}$ uses is lower bounded by $b \geq \log |\mathcal{H}|$, as the hypothesis in $\mathcal{H}$ returned by the algorithm must be computed from its internal state. Thus the memory requirement of the proper learning algorithm is $O(b)$ bits.

Now we can apply Proposition 5, as for large enough constant $M$, for $m \geq M$, the probability to succeed, $2/3$, is $\Omega(1/m)$. We get that $m = d^{\Omega(1)}$ or $b = \Omega(\log^2 d)$. Equivalently, $d = m^{O(1)}$ or $d = 2^{O(\sqrt{b})}$. In other words, $SQ_Q(\mathcal{C}) \leq \max(m^{O(1)}, 2^{O(\sqrt{b})})$.

$\square$

**Lemma 6.** *For any $\gamma \in [-1, 1]$ and $\tau \in (0, 1]$, there are $n = O(1/\tau)$ numbers $y_1, \ldots, y_n \in \{-1, 1\}$ such that $|\frac{1}{n} \sum_i y_i - \gamma| \leq \tau$.*

*Proof.* Take $n$ such that $1/n < \tau$. Let $k \in \{0, 1, \ldots, n\}$ be such that $(n - 2k)/n$ is $1/n$ close to $\gamma$. Take $y_1 = \ldots = y_k = -1$ and $y_{k+1} = \ldots = y_n = 1$. $\square$