[Reviews · NeurIPS 2020]

Review 1

Summary and Contributions: This paper describes a combinatorial property of a concept class C with respect to a distribution P, and proves that this property is intimately connected to bounded memory learning: when it is small, there exists a bounded-memory learner, and when there exists a bounded-memory learner then it is small’ (where small and small’ don’t exactly match in general but do match in certain interesting regimes). The property specifically is that C should have bounded SQ dimension with respect to any distribution Q that is pointwise multiplicatively close to P (“differential-privacy close”). Their results also show that this property is similarly intimately connected to strong SQ-learning, and thus they show that in interesting parameter regimes, bounded-memory learnability is equivalent to strong SQ learnability. This is particularly interesting because there exist examples that seem to separate the two, but it really gets at exactly how the quantities are defined. Overall, I find this paper to be a significant contribution to our understanding of learnability, and I advocate for its acceptance.

Strengths: Gives an interesting combinatorial property that provides insight into and sheds new light on memory-bounded learning, and learnability more generally. Relevant to current interests in the field.

Weaknesses: Doesn’t completely solve the problem.

Correctness: Yes

Clarity: Yes, quite well written.

Relation to Prior Work: Yes

Reproducibility: Yes

Additional Feedback: One question I have is: can you say more about any relationship between your characterization and that in [13] or [36] for strong SQ learning? These give different characterizations, and given your results I wonder if you can say something about how they fit together. Added: thanks to the authors for their response.


Review 2

Summary and Contributions: The paper gives the first characterization of the existence of a nontrivial bounded memory learning in terms of a combinatorial dimension. More precisely, the paper considers the task of learning a {-1, 1}-valued function c from a class C given m examples of the form (x, c(x)) where x is sampled from a fixed distribution P on a set X (exactly realizable, fixed distribution setup), while only being given b bits of memory. The paper focuses on the case when |C|, |X| = poly(N), in which case there are two trivial algorithms: (1) ERM uses m = O(log N) samples and b = O(log^2 N) bits; and (2) brute force over C uses m = poly(N) samples and b = O(log N) bits. The paper defines (P, C) to be “bounded-memory learnable” if it can be learned in a way that beats both (1) and (2) i.e. with m = N^{o(1)} and b = o(log^2 N). In this setting, it is shown that (P, C) is bounded memory learnable up to accuracy 1 - eps if and only if a robust notion of the SQ dimension of (P, C) is bounded by poly(1/eps). The paper’s theorems also apply more generally to other regimes |C| and |X| but the equivalence no longer is tight in these regimes. UPDATE: Thank you to the authors for the response and changes. My review remains that this paper should be accepted.

Strengths: The main strength of the paper is that this is the first characterization of bounded memory learning in terms of a combinatorial dimension, which is a conceptually important step in a hot research topic in learning theory. As mentioned in the paper, these kinds of characterizations are classical results in various models in statistical learning theory e.g. VC dimension for PAC learning, SQ dimension for the SQ model and Littlestone dimension for online learning. Bounded memory learning has received a lot of attention in the past few years since the seminal paper of Raz (2016) and is generally a technically difficult topic, with lower bounds that require difficult and involved arguments. Furthermore, the parameter regimes where the equivalence in the paper is not tight do not seem as conceptually important.

Weaknesses: The connection here between SQ dimension and bounded memory learning is not new and has appeared previously in the literature. Furthermore, while bounded memory learning is a technically difficult topic, the paper does not put forward any new algorithms or lower bounds for bounded memory learning. Rather, it leverages existing bounds: it uses Proposition 3 which appeared in Steinhardt et al. to show that SQ learnability implies the existence of bounded memory algorithms and Proposition 5 from Garg et al. to show that SQ dimension implies lower bounds for bounded memory learning. The main technical steps in the paper are boosting arguments to make these applications of Propositions 3 and 5 yield a tight equivalence. While these boosting arguments are nontrivial, from a technical perspective, this paper feels more like a nontrivial application of prior results than a set of new techniques. While the proof of Proposition 3 in Steinhardt et al. is not that involved, it seems as though the bounded memory lower bound in Proposition 5 is involved, and that the main result in the current paper leans on it heavily. However, I do not view these as serious drawbacks since the paper’s focus is not necessarily to introduce new techniques but rather to make a conceptually significant contribution by observing a tight characterization of bounded memory learning. The paper achieves this goal and I think should be accepted.

Correctness: The proofs appear to be correct and are well-explained. This is a purely theoretical paper that shows lower bounds and thus has no empirical component.

Clarity: The paper is very well written and organized. This is a strength of the paper. In particular, the figures illustrating proof outlines are very helpful.

Relation to Prior Work: Section 1.3 does a very good job of discussing how this work fits into the surrounding literature.

Reproducibility: Yes

Additional Feedback:


Review 3

Summary and Contributions: The paper shows connections between bounded-memory learning and SQ learning. It shows an upper bound that any class with small SQ dimension can be strong learned with with better memory or sample complexity than the trivial algorithms. It also shows a lower bound that if a class is learnable with small memory or sample complexity, then its SQ dimension is small. The paper shows that the upper and lower bounds match in a certain regime of parameters, and conjecture that they match more generally. To show these results the paper leverages previous results on SQ learning, memory bounded learning and boosting along with some new connections.

Strengths: The problem considered in the paper is interesting and should be of interest to the NeurIPS community. The question is a natural one: is there a clean characterization of memory-bounded learning, such as what we have for PAC learning? The connections makes progress on answering this question, and the connections that it makes are interesting.

Weaknesses: Before discussing the weaknesses, let me describe the high level proof outline of the main results, starting with the upper bound (Thm 1). A previous result by SVW16 shows that if a concept class if learnable with statistical queries then it is learnable with bounded memory. Classic SQ bounds imply that if the SQ dimension is bounded then the concept class is weakly learnable with statistical queries. What the paper does here is make the bridge between weak learning and strong learning for SQ, using boosting. This is Theorem 3 of the paper. Given Theorem 3, Theorem 1 essentially follows from the results of SVW16. For the lower bound (Thm 2), a previous result by GRT18 shows that if the SQ dimension is large then the function is not learnable with bounded memory. But this result is for proper learning (recovering the unknown function with any non-trivial probability). The authors extend this to show a lower bound for improper learning as well. One could say that SVW16 and GRT18 already sort of establish a connection between SQ and bounded memory learning and this is one possible weakness of the paper. However, the current paper fills various gaps, such as between weak and strong learning and improper vs. proper learning, and hence is interesting. It shows that the upper bound and lower bound are tight for a certain regime, and it formulates an interesting Conjecture (Conj 1) for the general case. Also I am curious that Thm 3 and 4 for SQ were not known before? They seem to be quite fundamental results about SQ.

Correctness: The paper appears to be technically solid.

Clarity: The paper is quite well-written. It does a good job of describing the high-level proof ideas. Fig. 1 and Fig. 2 are helpful to understand the proof structure.

Relation to Prior Work: The paper discusses prior work thoroughly and quite well.

Reproducibility: Yes

Additional Feedback: Some other points: ---Fig. 1 and Fig. 2 have been interchanged. ---Line 69: update -> updates ---It would be nice to see a bit more discussion about Conjecture 1 and its implications. ---Line 226: is has -> has ---Line 257: Proposition 6 : appears to be an incorrect reference ---Line 272 and 273: Citation should be for GRT18, instead is to GRT19. ---Maybe end with a conclusion and future directions. ------Post author response------- Based on the author response and discussion, I am satisfied that the connections to strong SQ are novel and interesting, and am hence updating my score.


Review 4

Summary and Contributions: The paper studies combinatorial characterizations of bounded memory learning which is an area which seen a recent surge of interest within the learning theory community. The paper builds on previous results on characterizing weak learnability of a concept class using bounded SQ dimension to come-up with a SQ dimension bound on the class under all distributions “close” to the original sampling distribution which characterizes (in certain regimes) strong learnability in PAC and SQ regimes. The authors build on a boosting algorithm to obtain their upper bounds. They provide complimentary lower bounds as well which in certain regimes are tight with the upper bounds.

Strengths: The claims of the paper are sound and written in a clear manner. The contribution is novel and studies a very relevant problem to the NeurIPS community. The paper achieves its goal of providing a general combinatorial characterization for bounded memory learnability under two well-studied models (PAC and SQ). The generality of the result is a positive aspect of the paper.

Weaknesses: The clarity of the paper is lacking in certain places and the presentation flow could be made better. The contribution of the paper is significant however it is unclear how novel it is in light of previous work existing on characterizing bounded memory learning. Perhaps some progress on answering Conjecture 1 would make the paper stronger, the reason being as follows: The notion of close distributions which have their SQ dimension bounded appears like an artificial construct created for proofs to go through. If this quantity is indeed the right quantity which characterizes bounded memory learning in all parameter regimes then it is a very significant quantity. Given the partial results in this paper, it is unclear if that is indeed the case.

Correctness: I have not verified all the claims. The ones I verified are correct and the rest of the paper appears sound too.


Clarity: The clarity of the paper is lacking in certain places and the presentation flow could be made better. The paper uses terms from the SQ model in many places in different ways and at times it can be confusing to a non-expert reader.

Relation to Prior Work: Yes

Reproducibility: Yes

Additional Feedback: Suggestions: 1. The claim in the abstract that this work is the first combinatorial characterization of bounded memory learning seems dubious given the prior work of [37] characterizing weak learnability. I would suggest making it more clear that this is first characterization for strong learnability as opposed to weak learnability.
 Questions: 1. What happens with infinite sized concept classes? Can we use covering number bounds instead of log |C|? Minor Typos: Line 43: Definition 3 is a broken link Line 219: “Similarly” -> “Similar” ----- Post author-response ----- Thank you to the authors for their response. The author response and discussion have changed my opinion of the paper slightly and I have updated my score accordingly.

[Author Response · NeurIPS 2020]

We thank the reviewers for their suggestions and positive feedback.

Reviewer #1, #3 - although both are notions of strong SQ, in the characterizations of [36,13] the distribution is fixed and the SQ-dim is computed over different classes. On the other hand, we fix the class and consider the SQ-dim over different distributions. This characterization was useful for us when relating it to bounded memory learning, which is the main focus of the paper.

Reviewer #4 -

- If the class is infinite, then most of the class hypotheses cannot be returned, as the memory is bounded. Therefore, the assumption that the class is finite is crucial. We know from [6] that distribution-dependent learning is equivalent to finite covering. So, as you wrote, one can focus on the cover of the class.

- By definition, small SQ-DIM implies weak bounded-memory learning (no need to use the general reduction from SQ to BM from [37]) since it suffices to find the correlation with a small number of hypotheses. The opposite direction is more challenging. Even though [17] is doing the heavy-lifting, there are still gaps even for weak learning, which we fill in this work.

- As Reviewer #2 pointed out, our contribution is conceptual; we introduce a new combinatorial definition and observe a tight characterization in a parameter regime. Obviously, it is desired to prove the characterization holds in the entire regime.



[Meta-Review · NeurIPS 2020]

The submission studies a novel approach to characterising learnability in an important learning model. In their initial reviews, some reviewers expressed minor concern about the technical novelty of the work. After the authors' reply and discussion, some reviewers raised their scores. The reviewers agree that the results are interesting and the paper should be published.